# Face Processing in Prematurely Born Individuals—A Systematic Review

**DOI:** 10.3390/brainsci14121168

**Published:** 2024-11-22

**Authors:** Tiffany Tang, Kasper Pledts, Matthijs Moerkerke, Stephanie Van der Donck, Bieke Bollen, Jean Steyaert, Kaat Alaerts, Els Ortibus, Gunnar Naulaers, Bart Boets

**Affiliations:** 1Center for Developmental Psychiatry, KU Leuven, Herestraat 49 ON5B bus 1029, 3000 Leuven, Belgiummatthijs.moerkerke@kuleuven.be (M.M.); stephanie.vanderdonck@kuleuven.be (S.V.d.D.); jean.steyaert@uzleuven.be (J.S.); bart.boets@kuleuven.be (B.B.); 2Department of Development and Regeneration, UZ Leuven, Herestraat 49, 3000 Leuven, Belgium; bieke.bollen@uzleuven.be (B.B.); els.ortibus@uzleuven.be (E.O.); gunnar.naulaers@uzleuven.be (G.N.); 3Child Psychiatry, UZ Leuven, Herestraat 49, 3000 Leuven, Belgium; 4Research Group for Neurorehabilitation, Department of Rehabilitation Sciences, KU Leuven, Tervuursevest 101 Room 02.57, 3001 Leuven, Belgium; kaat.alaerts@kuleuven.be

**Keywords:** preterm birth, face processing, social difficulties, psychopathology

## Abstract

Background/Objectives: Prematurely born individuals are at risk for developing socio-emotional difficulties and psychopathologies such as autism spectrum disorder. Particular difficulties processing social information conveyed by the face may underlie these vulnerabilities. Methods: This comprehensive review provides an overview of 27 studies published between 2000 and mid-2022 concerning face processing in individuals born preterm and/or born with low birth weight across different age ranges, paradigms, and outcome measures. The results were interpreted across different developmental stages. Results: Behavioural studies indicated that prematurity is associated with poorer facial identity and expression processing compared to term-born controls, especially for negative emotions. Structural alterations and delayed maturation in key neural face processing structures could explain these findings. Neuroimaging also revealed functional atypicalities, which may either be rooted in the structural alterations or may partly compensate for the delayed maturation. Conclusions: The results suggest that altered face processing may be associated with an increased risk of developing psychopathologies in individuals born prematurely. Future studies should investigate the preterm behavioural phenotype and the potential need for face processing rehabilitation programs.

## 1. Introduction

Fifteen million babies are born preterm worldwide every year, which is more than one in ten babies, and this number is rising [1]. Prematurity is typically defined based on gestational age (GA). Preterm (PT) birth refers to babies born before 37 weeks of pregnancy, with the subcategories of late preterm (LPT; born between 32–37 weeks), very preterm (VPT; born between 28–32 weeks), and extremely preterm (EPT; born before 28 weeks) [1]. Another way to characterize prematurity is based on birth weight (BW), as most PT babies are small and weigh less. Low BW (LBW) refers to babies weighing < 2500 g at birth, very low BW (VLBW) refers to babies weighing < 1500 g, and extremely low BW (ELBW) refers to babies weighing < 1000 g [2]. Complications of PT birth (e.g., respiratory distress syndrome, hypothermia, infections) are the leading cause of death among infants. As a result of important medical and technological advancements in neonatal intensive care units (NICUs), the survival rate of PT infants has improved significantly. However, PT birth survivors still face significant challenges throughout their lifespans, including cognitive, behavioural, social, and emotional problems [3]. Several studies have revealed that PT children are susceptible to developing symptoms of attention deficit hyperactivity disorder (ADHD), autism spectrum disorder (ASD), and anxiety disorders [4,5], rendering a comparably higher prevalence of these diagnoses in the population of PT children [6]. This led to the conceptualization of the “preterm behavioural phenotype”, an umbrella term referring to difficulties with attention, anxiety, emotion processing, socialization, and internalizing behaviour problems [7,8,9].

Montagna and Nosarti [9] have developed an integrative framework of this complex interplay among PT birth and socio-emotional vulnerabilities that may eventually translate into certain psychopathologies, with underlying biological and environmental factors. Biological factors such as structural and functional differences in the so-called “social brain” [10] can be associated with differences between PT and full-term (FT) children concerning social cognition [9]. The social brain is responsible for recognising facial expressions and facial identities, mentalizing, social competence, and social–emotional processing. Structures involved in these social processes include, but are not limited to, the amygdala, the medial prefrontal cortex, the corpus callosum, the cingulate cortex, and various regions within the temporal lobe [11]. PT birth may interrupt the typical maturation of these structures during the second and third trimesters of foetal development [10], thereby impacting brain activity and structural and functional connectivity [12]. For instance, PT infants show atypical neural activity and connectivity in frontotemporal regions while performing theory of mind (TOM) tasks that require making mental representations of mental states of others [13,14]. Another crucial skill for social communication is face processing as faces offer several important cues for social interaction by conveying information, such as identity, expression, mood, and gaze direction. Visual information about faces passes along two important neural pathways: (i) a subcortical system that engages in detecting faces and directing visual attention to them and (ii) a core cortical system for the detailed visual–perceptual analysis. Both of these pathways interact with (iii) an extended system involved in further processing of faces [15,16]. The subcortical system (including the retina, the superior colliculus, and the pulvinar) leads to visual preference for faces in newborns and plays a role in the specialization of the other cortical regions. The core cortical system consists primarily of the occipital face area and the fusiform face area, which use invariant facial features to process facial identity, and the posterior superior temporal sulcus (STS), which relies on more dynamic eye and lip movements to process facial expressions and eye gaze [11,17]. The extended system encompasses several regions (including the amygdala, the medial prefrontal cortex, the intraparietal sulcus, the anterior temporal pole, and the insula) and helps further processing, such as processing of emotional intentions and the appropriate social response [11,17]. Throughout childhood, these systems develop, and face processing generally evolves from an analytical or featural approach to a holistic approach, meaning that children initially mainly process isolated facial features and their spatial relations to each other and slowly grow to perceive these features as a whole [18].

In addition to brain alterations related to prematurity, early-life factors such as increased stress and pain (e.g., during the NICU stay) may also impact brain maturation or contribute to atypical family dynamics, potentially affecting socio-communicative and socio-emotional development in PT individuals [9,19]. Childhood adversity has been found to have a significant impact on facial expression processing in adolescence, indicated by reduced discrimination between angry (or threatening) and neutral (or safe) faces [20]. Similarly, the challenging NICU stay with many (painful) medical interventions could also affect facial expression processing in PT individuals. Besides contributing to difficulties in areas like face processing, these early-life factors may have profound repercussions on social development and the fluency of social interactions [21], increasing the risk of later (subclinical) psychopathology. Altered or impaired face processing has been reported in, and linked to, several psychiatric conditions [22,23,24,25,26,27]. Of particular interest is ASD, a neurodevelopmental disorder characterised by difficulties with social communication and interaction; the prevalence of ASD has been shown be to be increased by a factor of ten in the PT population, with a recent Swedish national cohort study showing that ASD prevalence was 6.1% for EPT, 2.6% for VPT, and 1.9% for LPT individuals [6].

Here, we aim to present a comprehensive review of face processing capabilities in PT individuals compared to term-born controls. This includes a qualitative analysis across various age ranges (from infants to adults), various paradigms, and various outcome measures (behavioural, eye tracking, and neuroimaging). Our review provides the first systematic overview of the evidence for facial identity and expression processing difficulties in PT individuals throughout the lifespan, and thereby complements previous reviews in PT individuals by Dean et al. [28] on (mainly non-social) visual attention up till five years of age and by Burstein et al. [29] on social cognition up till two years of age.

## 2. Materials and Methods

Eligibility criteria: There were few restrictions on the type of studies as we aimed for an inclusive review on face processing in PT individuals across the entire lifespan and various paradigms and measures. We included prospective and retrospective comparative cohort studies, case–control studies, and cross-sectional studies. Conference reports and poster abstracts were excluded. To be included, articles needed to be available in English, peer reviewed, and provide a behavioural outcome related to face processing performance or its structural or functional neural correlates. Participants consisted of individuals born prematurely, selected on the basis of GA, or born with low BW, or both, across various age ranges (from infants to adults). The majority of the studies also included FT individuals or individuals born with normal BW (NBW) as controls. Since both prematurity and the specific NICU experience influence face processing development in preterm birth survivors, we included only studies reflecting these modern NICU practices to ensure our conclusions are relevant to current clinical practice and research. Thus, studies published before 2000 were not included in this review.

Search strategy: This review was registered on PROSPERO on 24 March 2022 with registration number CRD42022302792. During the different stages of this review, the PRISMA checklist [30] was consulted. We conducted systemic electronic searches within PubMed (Medline), Embase, and Web of Science Core Collection on 18 May 2022. We also conducted a “snowball” search to identify additional studies by searching the reference lists of publications eligible for full-text review. To identify the target study population, we used several search terms, including “infant, premature”, “premature birth”, and variations of these terms. Considering low GA and BW, we included “infant, low birth weight”, “small for gestational age”, and variants of these terms. Various turns of phrases were used to define face processing, such as “face recognition”, “face perception”, “facial expression”, and variations of these terms. For a detailed overview of the search terms, see Appendix A. One author searched the databases and selected relevant studies with supervision and validation by a second author. In case of doubt or disagreement, a second opinion of another author was sought during consensus meetings.

Data collection process: A data extraction table was developed, evaluated using three articles, and then refined. After finalizing the table, one author performed the initial data extraction for all included articles and a second author checked all proceedings. Data abstracted included participant characteristics, methodology, key face processing results, conclusion and interpretation by the authors, and limitations, as mentioned in the articles.

Risk of bias assessment: Ultimately, 27 articles were retained for this review. For a detailed overview, see Appendix B. For the 13 articles involving cohort and case–control studies, the Newcastle–Ottawa Scale (NOS) was used to evaluate the quality of the study on three aspects: (i) the selection of study groups, (ii) the comparability of target and control groups, and (iii) the ascertainment of the exposure and outcome [31,32]. For the 14 other articles, we used the Joanna Briggs Institute (JBI) Critical Appraisal Checklist, as the literature favours this bias assessment tool for cross-sectional studies [31,33]. We interpreted prematurity or low BW as exposure, information on GA as ascertainment of exposure, and considered poor face processing as outcome of interest. Certain exclusion criteria (e.g., absence of neurological disease) and matching on developmental age, IQ, and biological sex were seen as essential factors that a study needed to take into account, because cognitive tasks are often used to evaluate face processing and sex differences in emotion processing performance [34]. In addition, as it may be theoretically and clinically relevant to disentangle the relative impact of GA versus BW, we considered BW as a confounding variable when the focus of the study was prematurity based on GA, and vice versa when the study focused on BW. Moreover, structural development or maturation of the core and extended cortical face processing systems continues into adulthood [17]. Therefore, an ideal or strict follow-up age was not defined as we wanted to investigate results through the whole development across various ages, ranging from infancy to adulthood.

Data analysis: This review aims to integrate several different outcome measures. Due to the large variability in outcome measures, a quantitative meta-analysis was not feasible. To systematically describe and evaluate all findings, the articles were grouped in terms of methodology, i.e., behavioural versus neuroimaging studies. The behavioural studies were further subdivided in facial identity or facial expression recognition tasks and eye tracking or visual attention orientation studies. The neuroimaging studies were grouped according to imaging technique, i.e., magnetic resonance imaging (MRI), electroencephalography (EEG), magnetoencephalography (MEG), and functional near-infrared spectroscopy (fNIRS). After reviewing the primary results, all articles were discussed chronologically throughout the different developmental stages.

## 3. Results

### 3.1. Literature Search

The search strategy (Figure 1) initially produced 252 results across the three consulted databases. Filtering out doubles resulted in 126 unique articles. After a first round of screening based on title and abstract, 26 articles remained. Four articles were excluded as no full text was available (e.g., poster abstracts). After reading the full texts of the remaining articles, two articles were excluded as they did not present general social cognition data as opposed to face processing data. Through the principle of “snowballing”, we found an additional seven articles that fitted our criteria. Ultimately, 27 articles were included for this comprehensive review.

### 3.2. Risk of Bias

In general, the quality of the search results was deemed satisfactory based on the following bias assessments.

#### 3.2.1. Case–Control or Cohort Studies (*n* = 13)

None of the case–control or cohort studies showed a high risk of bias (Table 1). The majority of studies (62% or *n =* 8) showed moderate risk of bias with the remaining studies (38% or *n* = 5) showing little risk of bias. Regarding participant selection, most of the studies used a representative study population. Most studies explicitly excluded participants with (a medical history of) brain lesions, neurological diseases, or abnormalities, with the exception of three studies that did not specify this in the description of their study populations. Six studies matched the PT and FT groups based on sex and corrected for IQ in the statistical analysis. Another three studies also matched the groups on sex yet lacked correction for IQ. Additionally, only one study examining GA controlled for BW, whereas none of the studies looking at BW controlled for GA. Seven articles studying infants or school-aged children followed the participants into adulthood.

#### 3.2.2. Cross-Sectional Studies (*n* = 14)

Most of the cross-sectional studies checked off at least half of the criteria on the JBI checklist (Table 2). They defined their sample properly and met the criteria used for measurement. Two studies did not exclude individuals with any history of brain lesions or neurodevelopmental disorders. Matching of participants based on sex, confounding factors, and strategies to deal with them is mentioned in five studies but only three of them discussed both sex and IQ level, whereas the remaining two only matched on sex. Only one cross-sectional study corrected for BW to investigate the relationship between GA and BW. One study did not use appropriate statistical analysis.

### 3.3. General Study Characteristics

Grouping the 27 articles in terms of methodology yielded 18 behavioural studies and 9 studies using neuroimaging techniques (Figure 2). Of the behavioural studies, five studies assessed visual attention orientation towards faces (four studies with and one study without implementing eye tracking) while the other 13 assessed performance on facial identity or facial expression processing tasks. Regarding the neuroimaging studies, three studies used MRI, two employed EEG, one made use of fNIRS, and three applied MEG.

Most studies investigated infants or toddlers (*n* = 10) or school-aged children (*n* = 7), four studies investigated adolescents, and six studies investigated young adults (Figure 3). Notably, PT neonates up till 6 months of age and PT children aged one-to-three years old seem to be under-analysed populations in both behavioural and imaging studies (Figure 3). Typically, congenital abnormalities, sensory impairment, and major brain lesions were incorporated as exclusion criteria. Most studies contrasted groups of children based on GA (*n* = 20), some studies focused on BW (*n* = 5), and a minority of studies explicitly disentangled the relative impact of GA versus BW (*n* = 2). All but one study applied a PT versus FT group comparison design, contrasting PT individuals with age-matched FT controls (for the infant studies, PT-corrected age was used as a matching criterion). The remaining study applied a correlational design, correlating individual differences in GA and BW with individual differences in face processing performance. In this review, most articles focused on prematurity defined by GA, although some studies did specifically target BW. Therefore, we will consequently characterize each of the study populations in the most comprehensive manner by referring to both GA and BW.

### 3.4. Behavioural Studies

#### 3.4.1. Facial Identity and Facial Expression Processing Performance

Most behavioural studies investigated either facial expression processing (*n* = 8) or facial identity recognition/memory (*n* = 4), while one study investigated both (*n* = 1) as part of a broad array of neurocognitive functions. Pertaining to facial expression processing, all nine studies (PT/LBW *n* = 1; VPT/VLBW *n* = 3; EPT/ELBW *n* = 5) found that PT individuals (toddlers *n* = 1; children *n* = 5; adolescents *n* = 1; adults *n* = 2) made more errors on free expression labelling tasks, indicating that they were less accurate in naming emotional expressions without any context. In contrast, two studies (PT/LBW children *n* = 1; EPT/ELBW toddlers *n* = 1) observed no difficulty in the (E)PT groups in choosing and recognising the appropriate facial expression for/in a given social context, suggesting that contextual information might help [35,36]. A study by Gao et al. [37] found no difference between ELBW/EPT and NBW/FT adults in the threshold to detect emotions in facial expressions that gradually increase in emotion intensity, indicating that the EPT group required the same level of intensity to detect emotions. Yet, a strong bias for fear was demonstrated as EPT/ELBW adults did misidentify angry faces more often as fearful ones, which might suggest a hypervigilance to potential threats [37]. A second study by Wocadlo and Rieger [38] also found that VPT/VLBW children made errors in recognizing angry faces and that their performance was positively associated with the abilities to initiate social interactions and to communicate, as assessed by the assertion and responsibility subscales of the parent- and teacher-reported Social Skill Rating System (SSRS). Next, besides poorer overall emotion recognition, O’Reilly et al. [4] found lower recognition for disgusted faces in EPT/ELBW adults compared to FT adults. Using the Child and Adolescent Social Perception Measure (CASP), Williamson and Jakobson [39] demonstrated that VPT/VLBW children had problems interpreting facial expressions due to difficulties in identifying face and body cues while relying more on situational cues and that this failure could lead to parent-reported social difficulties. Another study by Marleau et al. [40] stated that EPT/ELBW children scored poorer on a free facial expression labelling task and a TOM assessment battery compared to term-born controls; they also stated that individual differences in performance on these tasks could explain 45% of the variance in social functioning as assessed by the parent-reported Social Scale of the Adaptive Behaviour Assessment System—II. However, in contrast with the aforementioned studies using the SSRS, CASP, and TOM, Twilhaar et al. [41] did not find a direct association between the poorer facial expression processing in VPT/VLBW adolescents compared to term-born controls and the parent-reported social problems. Instead, deficits in cognitive control (e.g., difficulties reproducing spatial sequences of dots) were found to play a significant mediating role in the relation between VPT birth and parent-reported social problems [41].

Regarding face detection, facial identity recognition, and face memory, all five studies reported lower performance in PT individuals (PT/LBW children *n* = 1; VPT/LBW adolescents *n* = 1; EPT/ELBW children *n* = 2, EPT/ELBW adults *n* = 1) compared to FT controls. Pavlova et al. [42] used a Face-n-Food task, consisting of a set of food images presented as a coarse face scheme (i.e., resembling eyes and mouth), which were shown in a predetermined sequence from least to most resembling a face. This task revealed that PT adolescents reported a face impression at a later point in time in that sequence, indicating deficits in the sensitivity or tuning towards faces [42]. Mathewson et al. [43] implemented a visual discrimination task to evaluate the impact of visual configural processing, specifically the spacing among features of human faces, monkey faces, and houses. Feature spacing in the human and monkey faces was manipulated by changing the positions of the eyes and the mouth, while variations in feature spacing in the house images consisted of different positions of the upper and lower windows [43]. As EPT adults had difficulties exploiting differences in feature spacing to discriminate different identities among any of the three categories, this suggests a general problem with visual configural processing and not specifically with facial identity processing [43]. A study by Perez-Roche et al. [44] used the facial memory subtest of the Test of Memory and Learning (TOMAL) and observed poorer immediate facial identity recognition in PT children compared to FT children, but similar delayed facial identity recognition or face memory performance. However, those children who were small for GA showed poor immediate facial identity recognition as well as poor face memory performance [44]. The observation that BW has a more decisive impact on facial identity recognition [44] was also confirmed by a study using the Cambridge Face Memory Test [45]. Lastly, Potharst et al. [46] also found that 5-year-old EPT/ELBW children scored worse on immediate facial identity recognition and on facial emotion recognition compared to FT controls. Moreover, these authors demonstrated that bronchopulmonary dysplasia and low socio-economic status were associated with poorer performance on the face processing tasks and foreign parental country of birth with better performance [46].

#### 3.4.2. Visual Attention Orientation Towards Faces and Facial Features

Pereira et al. [47] presented whiteboards shaped as a head and neck to two-day-old PT/LBW infants to assess eye or head movements towards normal and distorted facial stimuli, which an examinator then evaluated. These infants demonstrated reduced orienting movements towards both normal and distorted faces, showing no preference between the two, whereas FT/NBW infants clearly preferred the normal face templates [47]. Next, four studies used a Tobii eye tracking device to investigate gaze patterns in eight–ten-month-old infants (PT/LBW *n* = 2; VPT/VLBW *n* = 1) and 5-year-old children (VPT/VLBW *n* = 1). Across various tasks, VPT/VLBW infants consistently showed shorter fixation times and reduced preference for socially relevant content, including faces, and had a lower preference for looking at the eyes than the mouth, as compared to FT/NBW infants [48]. Yet, a longitudinal study by Dean et al. [49] (including the VPT/VLBW participants of the study by Telford et al. [48]) did not find a significant group difference in social attentional preference in the same tasks in VPT/VLBW children at 5 years old. Next, when viewing a video of a talking woman, both PT and FT infants preferred looking at the eyes than the mouth [50]. Investigating the impact of the infants’ native and non-native language, Berdasco-Muñoz et al. [50] found that, while PTs had similar scanning patterns for both languages, FTs looked more towards the eyes than the mouth for the native language as opposed to the non-native language. Moreover, LBW infants, ranging from VPT to LPT, displayed shorter total gaze time on facial stimuli and less frequent attention shifts between different faces during the familiarization phase with initially unfamiliar facial stimuli compared to controls [51]. However, no significant group differences were observed in gaze time or attention shift frequency when novel faces were introduced alongside familiarised ones [51]. Altogether, these findings indicate that PT infants display distinct attentional processes and face processing patterns, suggesting a different social orienting profile compared to FT infants.

#### 3.4.3. Key Findings of Behavioural Studies

PT individuals demonstrate poorer facial expression processing, particularly for angry faces and in the absence of contextual information. They also show difficulties in face detection and facial identity recognition, possibly due to a more general weakness in configural processing. PT infants show less orientation towards and preference for faces.

### 3.5. Neuroimaging Studies

#### 3.5.1. Studies Using (f)MRI

One study used structural magnetic resonance imaging (sMRI) in VPT/VLBW adolescents, a second study functional MRI (fMRI) in VPT/VLBW adults, and a third study applied both techniques in EPT/ELBW adolescents. Using voxel-based morphometry, the sMRI study by Healy et al. [52] revealed increased grey matter volume in bilateral fusiform gyrus (FG) in a subgroup of VPT/VLBW adolescents, identified as “socially immature” on the social problems scale of the Child Behaviour Checklist compared to FT controls. Notably, VPT/VLBW adolescents were more frequently classified as “socially immature” than their FT peers. Moreover, in these “socially immature” VPT/VLBW adolescents, individual differences in left FG volume were more strongly related to individual differences in ipsilateral orbitofrontal cortex (OFC) volume, whilst this relationship was not observed in VPT/VLBW peers not classified as “socially immature” [52]. The authors speculated that larger FG and left OFC volumes in these individuals may be precursors or consequences of subclinical psychiatric problems such as social difficulties [52]. A later study by Grannis et al. [53] combined sMRI and fMRI and demonstrated reduced grey matter density in right FG as well as greater face-evoked activity in right FG in EPT/ELBW adolescents as compared to FT/NBW peers. Machine learning allowed researchers to assign group status based on right FG volume or activation with accuracy scores of 88.64% and 77.27%, respectively [53]. Combining the indices of right FG volume and activity increases the accuracy score to 95.45%, further demonstrating that EPT/ELBW youth show atypical structural and functional face processing mechanisms compared to their term-born peers [53]. Papini et al. [54] administered resting-state fMRI and a facial expression processing task and found a positive association in FT/NBW adults among functional connectivity between amygdala and posterior cingulate cortex and facial emotion recognition performance (in particular, the labelling of angry faces). Yet, in VPT/VLBW adults, this association was not present, suggesting particular difficulties processing angry faces in the PT group [54].

#### 3.5.2. Studies Using EEG

Two studies (ELBW adults *n* = 1; PT/LBW infants *n* = 1) used spectral power electroencephalography (EEG) in relation to face processing. Spectral power analysis assumes that the EEG waveform is a linear combination of oscillations at particular frequencies (typically characterized in terms of the delta, theta, alpha, beta, and gamma frequency bands). By decomposing the spectrum, the magnitude or power of each of these frequency components can be determined [55]. In an EEG study by Carbajal-Valenzuela et al. [19], PT/LBW infants showed higher absolute power in frontal regions during a facial expression processing task, while FT/NBW infants showed higher power in occipital regions. The authors suggested that increased frontal power in PT/LBW infants may reflect attention allocation efforts that fail to improve facial expression processing, whereas higher occipital power in FT/NBW infants likely indicates a more efficient allocation of sensorial and perceptual resources. Moreover, the EEG profiles of the PT/LBW group lacked the distinction between positive, negative, and neutral facial expressions observed in the FT/NBW group, suggesting altered and less specific facial expression processing in PT infants [19]. Another EEG study by Amani et al. [56] found that ELBW adults exhibited greater relative right frontal alpha asymmetry (FAA), an index associated with avoidance, withdrawal behaviour, and negative emotional responses. Indeed, ELBW adults showed a stronger relation between right FAA at 22–26 years old and avoidance of threat related stimuli (i.e., angry faces) in a dot probe task at 30–35 years, compared to the NBW adults [56].

#### 3.5.3. Study Using fNIRS

Frie et al. [57] applied functional near-infrared spectroscopy (fNIRS), which measures the concentration changes of oxygenated (HbO_2_) and deoxygenated (HHb) haemoglobin in brain regions, to investigate hemodynamic responses in EPT/ELBW infants when viewing faces. While FT/NBW infants showed selectively higher activity in right frontotemporal areas in response to their mother’s face as compared to faces of strangers, EPT/ELBW infants’ brains showed no preference for their mothers’ faces [57]. As activation of these areas is likely related to facial discrimination and recognition processes [58], this suggests difficulties in facial identity processing in EPT/ELBW infants. These authors also examined regional volume of FG and amygdala via sMRI and its correlation with face-evoked fNIRS responses in the EPT/ELBW infants. While no association was found during the presentation of the mothers’ faces, they observed a negative correlation between fNIRS activity and FG and amygdala volume when the EPT/ELBW infants looked at unknown faces [58]. The association among reduced regional volumes and enhanced neural activity echoes the sMRI and fMRI findings of Grannis et al. [53] and may represent a functional mechanism to compensate for structural alterations. For instance, as the fNIRS probe covers a large area of the frontotemporal region, the higher response might reflect the recruitment of additional regions to compensate for regions with atrophy when performing a challenging unfamiliar face processing task [57].

**Figure 2 brainsci-14-01168-f002:**
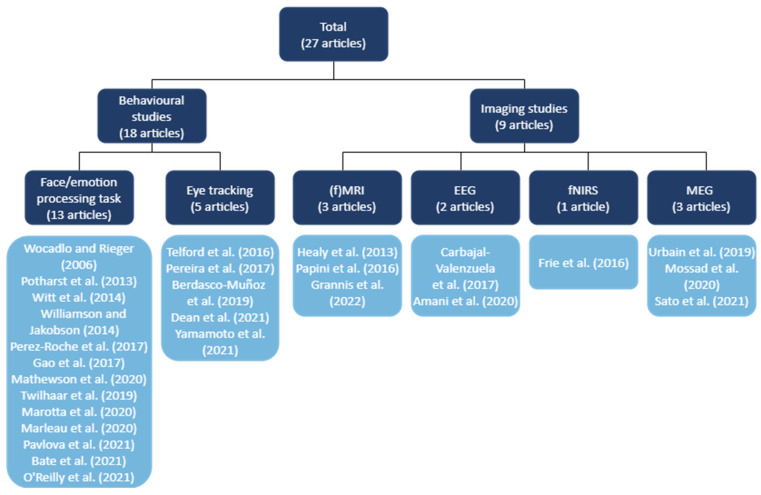
Schematic overview of the final selection of articles for inclusion in the synthesis, subdivided based on methodology for systematic analysis. Abbreviations: (f)MRI, (functional) magnetic resonance imaging; EEG, electroencephalography; fNIRS, functional near-infrared spectroscopy; MEG, magnetoencephalography. Articles referenced in the figure: [4,19,35,36,37,38,39,40,41,42,43,44,45,46,47,48,49,50,51,52,53,54,55,56,57,59,60,61].

**Figure 3 brainsci-14-01168-f003:**
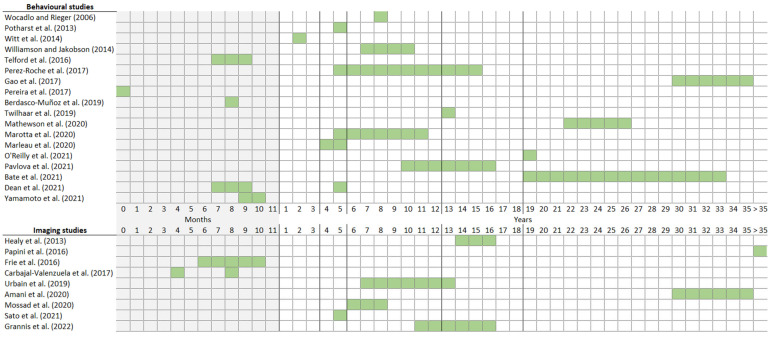
Schematic visualisation of the investigated age ranges (illustrated by the green boxes) of the included articles, subdivided by methodology (i.e., behavioural studies and imaging studies). Articles referenced in the figure: [4,19,35,36,37,38,39,40,41,42,43,44,45,46,47,48,49,50,51,52,53,54,55,56,57,59,60,61].

In the following paragraphs, studies will be grouped and discussed according to study paradigm and method to facilitate readability and insight.

**Table 1 brainsci-14-01168-t001:** Newcastle–Ottawa Scale per article to assess bias in the case–control or cohort studies.

	Number of Stars			
Study, Year	Selection *	Comparability ^×^	Outcome °	Score ˘
*Wocadlo and Rieger [38]*	3	1	3	7/9
*Potharst et al. [46]*	3	1	3	7/9
*Witt et al. [36]*	2	1	3	6/9
*Perez-Roche et al. [44]*	3	2	3	8/9
*Gao et al. [37]*	3	0	3	6/9
*Mathewson et al. [43]*	3	0	2	5/9
*Marotta et al. [35]*	3	1	2	6/9
*Healy et al. [52]*	3	1	3	7/9
*Carbajal-Valenzuela et al. [19]*	3	0	3	6/9
*Amani et al. [56]*	3	0	2	5/9
*Mossad et al. [60]*	2	0	3	5/9
*Dean et al. [49]*	3	1	2	6/9
*O’Reilly et al. [4]*	3	1	3	7/9

* Maximum number of stars for selection: 4 stars; ^×^ maximum stars for comparability: 2 stars; ° maximum number of stars for outcome: 3 stars; ˘ score of 7–9/9 stands for little risk of bias, while scores of 4–6/9 and 0–3/9 stand for moderate risk and high risk of bias, respectively.

**Table 2 brainsci-14-01168-t002:** JBI Critical Appraisal Checklist per article to assess bias in analytical cross-sectional studies.

	Total				
Study, Year	Yes *	No *	Unclear *	N. A. ^×^	Overall Appraisal °
*Williamson and Jakobson [39]*	7	1	0	0	Include
*Pereira et al. [47]*	4	4	0	0	Seek further info
*Twilhaar et al. [41]*	5	3	0	0	Include
*Marleau et al. [40]*	8	0	0	0	Include
*Pavlova et al. [42]*	5	3	0	0	Include
*Bate et al. [45]*	6	2	0	0	Include
*Telford et al. [48]*	6	2	0	0	Include
*Yamamoto et al. [51]*	5	2	1	0	Include
*Papini et al. [54]*	8	0	0	0	Include
*Frie et al. [57]*	5	3	0	0	Include
*Urbain et al. [59]*	8	0	0	0	Include
*Sato et al. [61]*	6	2	0	0	Include
*Grannis et al. [53]*	8	0	0	0	Include
*Berdasco-Muñoz et al. [50]*	6	2	0	0	Include

* Number of times the study scored positive (“yes”), negative (“no”), or “unclear” on an item on the checklist; ^×^ number of times an item on the checklist was not applicable for that study; ° the overall appraisal varies between “include”, “seek further info”, and “exclude”. Abbreviations: N.A., not applicable.

#### 3.5.4. Studies Using MEG

Three studies applied magnetoencephalography (MEG) during implicit facial expression processing tasks, in which participants had to press a button when a predefined visual cue (i.e., a car stimulus, a jumbled unidentifiable image, or a coloured frame around the faces) appeared on a screen, while expressive faces were presented. The study by Urbain et al. [59] examined brain activity in VPT children and correlated this with sMRI findings, while the other two investigated interregional brain connectivity in VPT/VLBW children and used sMRI to localise network hubs [60,61]. Urbain et al. [59] observed reduced activity in the right lateralized parieto–fronto–temporal network (including the postcentral, inferior temporal and angular gyri, the ventral prefrontal cortex, and the anterior temporal lobe) in VPT school-aged children compared to matched FT controls in the context of viewing angry faces. This reduced activity correlated with reduced cortical thickness in medial and ventral prefrontal cortex as assessed by voxel-based morphometry (sMRI) and was interpreted as evidence for altered facial expression processing [59]. Next, the study by Mossad et al. [60] administered a free labelling test of facial expressions (i.e., happy, angry, and fearful faces) and showed that 8-year-old VPT/VLBW children rated angry faces more positively compared to FT controls. The authors associated this behavioural performance with reduced connectivity in a theta band network assessed via MEG, comprising bilateral OFC, right amygdala, right FG, and right thalamus (as localised with sMRI). As negative emotions may be preferentially processed in the right hemisphere, the authors interpreted this reduced right-lateralized connectivity as an explanation for the reduced behavioural discrimination performance for angry faces [60]. Notably, although behavioural discrimination performance for fearful faces did not significantly differ between VPT and FT children, the association with reduced right-lateralized connectivity was also demonstrated for fearful faces [60]. Lastly, Sato et al. [61] found that VPT/VLBW preschoolers showed poorer affect recognition during a free labelling task compared to their FT peers, although parent-reported social and emotional behavioural functioning was similar in both groups. Using MEG, Sato et al. [61] identified decreased theta network connectivity during angry face processing, which sMRI anchored in left OFC, in VPT/VLBW preschoolers compared to FT controls. As the OFC region handles emotion processing and inhibitory control, this may relate to difficulties in negative affect facial expression recognition as well as difficulties in more general socio-emotional behaviour [61].

#### 3.5.5. Key Findings of Neuroimaging Studies

Altogether, MRI research revealed differences in the face processing network in PT individuals, not only at a structural but also at a functional level. EEG studies suggested that atypical attention allocation in PT individuals underlies altered sensitivity towards facial expressions and/or poorer performance in facial expression processing tasks. Combining MRI and fNIRS revealed compensatory overactivity in aberrant cortical face processing regions in EPT/ELBW infants, and a lacking neural differentiation between mother and stranger faces. Finally, based on MEG studies, reduced activity of the parieto–fronto–temporal network and decreased theta band connectivity have been put forward as biological markers of a less efficient facial expression processing system in PT individuals.

## 4. Discussion

This systematic review provides insights into potential differences in facial identity and facial expression processing between PT individuals and their FT peers. So far, we have presented the results per method of investigation. However, as several articles target different age groups and refer to the developmental aspect of face processing, we further discuss the results from a developmental perspective by relating them to the corresponding developmental stages.

### 4.1. Preterm Individuals’ Face Processing Abilities Throughout Development

During the first stage of life, FT infants develop face processing abilities to a great extent. Within hours after birth, newborns show an innate preference for human faces and clear distinction between human faces and other visual stimuli [17]. A looking preference towards the eyes as opposed to other facial features emerges at three months of age [17,62], although visual cues from the mouth also play a distinctive role in speech perception (e.g., the McGurk effect) [63]. Three–five-month-old FT infants recognise facial identities in an upright position using specific facial features and their spatial arrangements [17], termed as first-order configural processing [18]. In contrast, behavioural studies in PT infants found reduced orientation movements towards faces and no clear preference for faces [47]. Looking time at the eyes in faces was also reduced in PT infants, which is commonly associated with ASD characteristics [48,64]. Furthermore, group differences in total looking time at faces, eye gaze rate, and attention shifts suggest different attentional processes and facial cognition in PT individuals at an early age [51]. This may relate to cortical activation differences, as PT infants showed increased frontal power when looking at faces in an EEG study [19] and no distinction between unfamiliar and familiar (i.e., mother) faces was found in the hemodynamic responses during fNIRS imaging in PT infants [57]. A negative correlation between fronto–temporal activity measured by fNIRS and regional volumes investigated with sMRI (i.e., FG and amygdala) suggests a potential compensatory mechanism as activation of additional regions results in an oxygenated haemoglobin increase to substitute for the loss of cortical mass. This atypical mechanism might also partly explain reduced preferential orienting towards faces, as the amygdala is part of the subcortical attention-orienting system. In conclusion, premature birth seems to alter the development of social cognition and facial identity recognition during the first years of life.

The preoperational stage [65] reflects a transition point in psychological development, as toddlers and preschoolers become far more sociable and devote more resources to unfamiliar faces, as evidenced by greater neural responses to a stranger’s face as compared to their mother’s face [66]. Initially, a featural or analytical face processing approach is applied with a special focus on isolated face components such as eyes, nose, and mouth [18]. Subsequently, the development of a more configural approach, using second-order configurations involving the spacing of facial features relative to each other, further enhances facial identity recognition rapidly [67,68]. At 3.5 years of age, FT toddlers start perceiving emotions in a categorical manner (e.g., along a happy–sad continuum [69]), enabling the labelling of basic emotions. In contrast, PT toddlers show difficulties in recognizing and labelling facial expressions, which might relate to poorer performance on higher-level social cognition TOM tasks [36,40,46]. These difficulties may stem from structural and functional alterations of the FG, embedded in a core system supporting face detection and individuation [61,70]. This face-specific cortical vulnerability may also explain the basic social cognitive difficulties and need for additional contextual clues to understand social situations found in PT toddlers [36].

During the concrete operational stage [65] of school-aged children, individual facial features and their spatial arrangements are processed as an integrated whole [18,71]. Sensitivity to these spatial featural relations increases as the integrated core face processing network is forming [72,73]. This is reflected by the development of the N170 component, a negative deflection over occipitotemporal electrodes measured with EEG, reflecting face detection [74] and by reports of greater FG activation in fMRI studies [75,76]. In addition, the extended face network expands through task-specific selective recruitment [77]. Notably, MEG studies in PT children found reduced right hemisphere activity and functional connectivity among a broader face processing network during negative expression processing [59,60]. These findings may corroborate persisting difficulties in recognition of nonverbal cues and negative emotion and a continued need for additional situational social cues in PT individuals throughout childhood [38,39].

Both the core and extended face network mature further in typically developing adolescents (12–18 years) in a protracted way [17,78]. Contrastingly, PT adolescents show a delayed maturation of those networks, especially the right fusiform area [53]. Although compensatory hyperactivity in the FG was found with fMRI images [51], sMRI scans revealed larger volume and less dense grey matter of that region [53]. This altered structural maturation may result in delayed development of a coarse face scheme, as demonstrated with the Face-n-Food task, potentially underlying the facial identity and expression processing difficulties in PT adolescents [41,42].

Adult PT birth survivors show poor negatively valued facial expression processing [37], as reflected by greater relative right FAA at rest which predicted avoidance of angry faces and could result from stress-induced physiological changes due to PT birth [56]. More general visual configuration discrimination difficulties are also identified in PT adults, as poor discrimination performances were found for various stimulus categories with different feature spacing (i.e., monkey faces, human faces, and houses) [43].

Hypoconnectivity among a circuit managing emotional evaluation (including praecuneus, posterior cingulate cortex, and amygdala), as well as a potentially compensatory hyperconnectivity between amygdala (extended system) and STS (core system), were found in a resting-state functional connectivity study in PT individuals [54]. This combined observation of hypoconnectivity among the broader face processing network and hyperconnectivity among amygdala and face-sensitive regions may echo similar findings in adults with ASD [79]. Altogether, these findings might explain the longer reaction times in facial identity and expression processing of PT adults.

### 4.2. Structural, Functional, and Behavioural Face Processing Alterations in Preterm Individuals

Structures of the core and extended face processing system as well as related subcortical structures, in particular FG, OFC, and amygdala, develop differently in PT individuals. These structural alterations may be due to increased stress and pain related to preterm birth and the NICU stay during early life [19,56], as the NICU environment contrasts greatly with non-threatening intrauterine life [80]. Likewise, functional atypicalities in activity and connectivity of a parieto–fronto–temporal network and a theta band network (including bilateral OFC, right amygdala, right FG, and right thalamus) have been described, which may be rooted in the structural alterations or be a compensatory mechanism for the delayed neuroanatomical maturation. Across all reviewed studies, multiple explanations are offered on how these structural and functional alterations may result in behavioural differences between PT and FT individuals in terms of face processing ability and style. At a behavioural level, important differences in attentional processes and face perception between PT and FT individuals were identified, with reports of reduced orientation towards faces or socially relevant content and shorter fixation times when viewing faces in PT individuals [47,48,51]. Different social orienting profiles could lie at the basis of poorer performance in facial identity and facial expression recognition and labelling in the PT population, indicated by a higher error rate and longer reaction times compared to FT peers. Specifically for negative facial expressions, worse performances as well as a bias towards fear have been reported [37].

### 4.3. Birth Weight and Not Gestational Age Might Be a Better Predictor for Atypical Face Processing

Most of the reviewed studies selected a preterm study population defined by low GA and contrasted them with FT peers. As babies who are born earlier typically weigh less, BW and GA are closely related to each other [81], with a large overlap among the classical subcategories (i.e., ELBW and EPT; VLBW and VPT). Thus, it is not surprising that many authors did not make a distinction between the two factors. However, BW and GA are not perfectly correlated, as evidenced by the subgroup of low-for-gestational-age individuals. In this regard, it is striking that facial identity and expression processing performances strongly correlated with BW and birthweight-for-gestational age centile scores, but not with GA alone [45]. Similarly in another study, after correcting for BW, GA was no longer significantly associated with poor facial identity and expression task results [44]. Although these findings suggest that BW might be a better predictor for face processing performance in PT individuals, this must be interpreted against the background that BW and GA are not entirely independent variables of preterm birth.

### 4.4. Challenges, Limitations, and Opportunities in Preterm Face Processing Research

The search strategy for this systematic review yielded 27 useful results. Though it should be noted that the utilised search strategy might have missed some studies in the interval between 2000 and mid-2022, which should be resolved in future studies.

Reflecting on the reviewed studies, the following pitfalls and challenges can be identified. First, study design and setup varied greatly across studies and quality was slightly disputed. Bias was assessed by one author and was generally satisfactory.

Second, in the absence of a gold-standard face processing task or assessment, facial identity and expression processing was measured by multiple techniques and methods, including behavioural studies (i.e., explicit face processing tasks or eye-tracking) and neuroimaging studies (i.e., structural and functional MRI, EEG, MEG, and fNIRS). Although the multitude of approaches and outcome measures offers a broad and complementary overview of (aspects of) face processing in the PT population, it should be noted that different imaging techniques have their own advantages and disadvantages. For instance, pertaining to spatial and temporal resolution, EEG/MEG has high temporal resolution but limited spatial resolution, while MRI has high spatial resolution but limited temporal resolution. Moreover, different types of MRI scans investigate different aspects of the brain, with sMRI and fMRI examining brain structure and brain function, respectively. Therefore, when integrating findings across studies using different imaging modalities as opposed to a study that combines multiple imaging modalities, caution is needed as the imaging modality in combination with study-specific protocols and conditions has an undeniable impact on the results.

Third, most studies included a wide age range of participants but had a relatively short follow-up period. Thus, while developmental face processing aspects are frequently mentioned, this is mainly based on cross-sectional studies.

Fourth, the intrinsic heterogeneity that is embedded within the PT population is reflected by the definition of prematurity (based on GA or BW or both) among others, as the corresponding inclusion criterion varied greatly across studies and most studies did not account for both factors in the analyses. Moreover, the causes or mechanisms of preterm birth can vary extensively (e.g., malnutrition, traumatic events, medical complications, a multiple pregnancy) and could have a differential impact on later-life outcomes. Whereas GA is a clear indicator of preterm birth, low BW by itself might be linked to other underlying causes. Although most studies selected participants based on GA or reported both GA and BW, caution is warranted with studies defining their study populations solely by BW without additional information, as this might not accurately represent the preterm population. Other external factors such as neonatal patient characteristics, general cognitive abilities, and sociodemographic characteristics (e.g., socio-economic status) should also be documented and examined thoroughly as they might influence certain findings. Of importance are ethnicity and biological sex. While most studies included in the review did report on biological sex, the ethnicity of study participants was rarely elucidated. As Caucasian facial stimuli are often used and most studies took place in North America and Europe, this might introduce bias or underrepresentation in the gathered evidence of this review. Pertaining to studies on neonates, more details on NICU or hospital stay and medical condition (including the presence of medical devices or feeding tubes) should be given, as this could confound (neuroimaging) results and hinder integration of findings.

Fifth, not all studies provided a substantiated explanation for altered face processing in the PT populations. It is important to integrate findings in a more elaborated conceptual and theoretical PT face processing framework.

Sixth, while the current review focussed on facial expression perception (i.e., processing others’ facial expressions) in PT populations, it should be noted that facial expression production (i.e., posing certain facial expressions) is distinctive from, but closely related to, facial expression perception. Indeed, it has been well-established that facial expression perception is also influenced by proprioceptive feedback from our own facial muscles, especially when perceived expressions are ambiguous (e.g., [82,83]). In this context and given the enhanced experience of neonatal pain in the NICU, it is plausible that these earlier painful grimaces and the resulting facial muscle activity in PT individuals may have influenced their face processing development. Thus far, the reported studies did not assess the presence of certain facial expressions in the participants during the experiments, nor did they relate facial processing performance to neonatal pain or to the frequency of skin-breaking procedures at the NICU. However, future studies should aim to integrate both concepts and examine both expression perception and expression production, as well as their interdependencies and associations with early aversive experiences in an NICU (cf. [23]).

Seventh, though study sample sizes were generally substantial, many authors mentioned a potential bias due to modest sample size and a wish for larger study samples. Thus, to fully understand the development of face processing (difficulties) in PT individuals, well-powered and preferentially longitudinal cohort studies should be conducted, following infants throughout childhood and adolescence. 

Finally, PT children are at risk for developing psychopathologies like ADHD or ASD [3], which are often characterised by atypical facial identity and expression processing [22,24,25,26,27,84]. Here, throughout the present review, we demonstrate that PT individuals, in general, also show difficulties in face processing. Accordingly, a key question concerns the role of these face processing difficulties in the preterm behavioural phenotype and related psychopathologies. In other words, do our findings indicate that altered face processing puts PT individuals at risk for developing psychological disorders such as ADHD or ASD? Or is the observation of face processing difficulties in PT populations merely a consequence of their overlap with these other psychopathologies~ Tackling this question will entail a detailed analysis of individual profiles and subgroups of PT individuals. Further investigation is needed to better understand this (sub)clinical phenotype of prematurity and to examine possible rehabilitation strategies, as our results could have implications for clinical practice. More specifically, future studies could investigate the potential therapeutic role of facial identity and expression processing rehabilitation programs (e.g., [85]).

## 5. Conclusions

This review aimed to provide an overview of facial identity and expression processing in PT individuals. Correspondingly, we collected data from behavioural and neuroimaging studies to evaluate face processing in PT and FT individuals. Prematurity affects face processing throughout life as seen in the different developmental stages ranging from infancy to adulthood. In general, PT individuals showed difficulties in processing facial identities and expressions, particularly with negative expressions, compared to their FT peers. However, performance on both types of face processing can be partly compensated by additional contextual information. Structural alterations in key neural structures related to face processing, potentially resulting from early-life stress, may account for these difficulties. Functional differences may either be a consequence of these structural alterations or represent compensatory adaptations for delayed neuroanatomical maturation.

Future studies should further examine the effects of GA—or more specifically BW—on face processing across different age and weight groups with larger samples and evaluate the potential of rehabilitation programs for facial identity and expression processing. A quantitative meta-analysis would be valuable, though this requires optimized study designs, detailed reporting of study populations, and consistent outcome measures.

## Figures and Tables

**Figure 1 brainsci-14-01168-f001:**
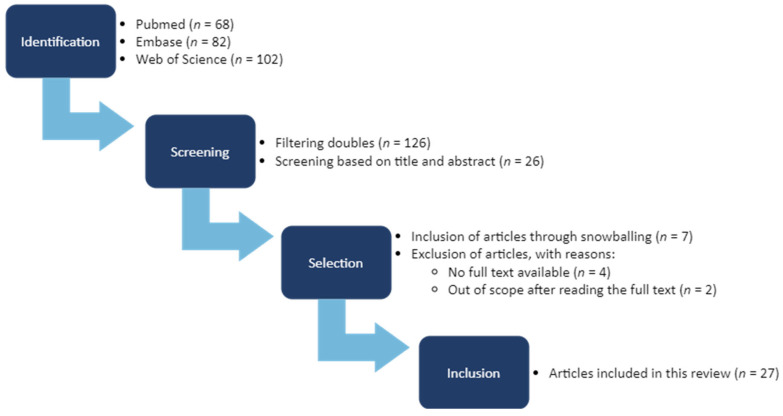
Schematic overview of the strategy from identification of articles in the databases to inclusion in the synthesis, according to the PRISMA checklist [30]. Abbreviations: *n*, the number of articles.

## Data Availability

No new data were created or analysed in this study. Data sharing is not applicable to this article.

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
