# Peer review of "Face Processing in Prematurely Born Individuals—A Systematic Review"

_brainsci, 2024, doi:10.3390/brainsci14121168_

Round 1
Reviewer 1 Report
Comments and Suggestions for Authors
This timely review by Tang et al. is an important contribution concerning with links between prematurity and neurobehavioral “social cognition” consequences including face recognition/processing and extended-ADHD-related symptoms.
The authors had checked most critical science-related databases for relevant studies in a proper way, avoiding the Google Scholar only.
All studies were classified and validated extensively, using the Newcastle-Ottawa Scale and the JBI Critical Appraisal Checklist.
Some improvements are possible for this manuscript.
[1]
Since low birth weight might be indicative for conditions other than pretermness, please indicate this possibility in the Discussion section and discuss extensively.
In particular, please discuss possible impacts of hypoxia-related antenatal exposure on babies’ low birth weight (high altitude pregnancies, prenatal hypoxia, etc), as well as possible impacts of malnutrition. If needed, please indicate the possibility to discriminate between these conditions in the studies analysed and a possible limitation for this review.
[2]
In the subsection “3.2.1. Case-control or cohort studies”, please finalize the subsection with summarizing sentences on % of studies with liitle, moderate and high risk of bias respectively.
[3]
Within the text of the manuscript, some papers are cited with irrelevant years
As a single example
Yamamoto et al. (2021), not Yamamoto et al. (2020)
Please correct.
[4]
Please check the eligibility of these studies for the review conducted:
Lundequist et al. Acta paediatrica (Oslo, Norway : 1992), 104(3), 292–299. https://doi.org/10.1111/apa.12864
Rose, S. A. (1980). Developmental Psychology, 16(2), 85–92. https://doi.org/10.1037/0012-1649.16.2.85
Field et al. (1983). Discrimination and imitation of facial expressions by term and preterm neonates. Infant Behavior and Development, 6(4), 485-489. (This item is available via Google Scholar)
Berdasco-Muñoz et al. (2019). Visual scanning of a talking face in preterm and full-term infants. Developmental Psychology, 55(7), 1353–1361. https://doi.org/10.1037/dev0000737
Pickens et al. (1994). Full-term and preterm infants' perception of face-voice synchrony. Infant Behavior & Development, 17(4), 447–455. https://doi.org/10.1016/0163-6383(94)90036-1
Harel et al. (2011). Gaze Behaviors of Preterm and Full-Term Infants in Nonsocial and Social Contexts of Increasing Dynamics: Visual Recognition, Attention Regulation, and Gaze Synchrony. Infancy : the official journal of the International Society on Infant Studies, 16(1), 69–90. https://doi.org/10.1111/j.1532-7078.2010.00037.x
Bova et al. (2022). Emotion recognition and theory of mind weakness at school age in children born preterm. Minerva pediatrics, 10.23736/S2724-5276.22.06581-8. Advance online publication. https://doi.org/10.23736/S2724-5276.22.06581-8
Butti et al. (2020). Premature birth affects visual body representation and body schema in preterm children. Brain and cognition, 145, 105612. https://doi.org/10.1016/j.bandc.2020.105612
Dean et al. (2021). Eye-tracking for longitudinal assessment of social cognition in children born preterm. Journal of child psychology and psychiatry, and allied disciplines, 62(4), 470–480. https://doi.org/10.1111/jcpp.13304
Wocadlo, Rieger (2006) Social skills and nonverbal decoding of emotions in very preterm children at early school age, European Journal of Developmental Psychology, 3:1, 48-70 https://doi.org/10.1080/17405620500361894
O'Reilly et al. (2021). Extremely preterm birth and autistic traits in young adulthood: the EPICure study. Molecular autism, 12(1), 30. https://doi.org/10.1186/s13229-021-00414-0
[5]
In the Discussion section, please identify uniqueness of this review vs Burstein et al. (2021) and Dean et al. (2021). works:
Burstein et al. (2021). Preterm Birth and the Development of Visual Attention During the First 2 Years of Life: A Systematic Review and Meta-analysis. JAMA network open, 4(3), e213687. https://doi.org/10.1001/jamanetworkopen.2021.3687
Dean et al. (2021). Social cognition following preterm birth: A systematic review. Neuroscience and biobehavioral reviews, 124, 151–167. https://doi.org/10.1016/j.neubiorev.2021.01.006
Please also consider comparing with other reviews in the field.
[6]
Please consider to do meta-analysis based on the data collected.
[7]
Please consider to extend the study with 2022-2024 yy manuscripts
Suggestions for future studies.
[S1]
At the age of perfected machine translation via Google or similar tools, is it optimal to exclude articles available in other languages?
Similar concerns are actual for preprints’ exclusions.
[S2]
In future studies, please consider to automatize relevant manuscript retrieval from databases with LLM-based systems like LLAMA or ChatGPT O1.
[S3]
Sometimes, a simpler search is more fruitful.
For example, “Preterm” AND “face recognition” in Google Scholar is effective.
Comments on the Quality of English Language
[L]
Some language improvements are required.
[L1]
A number of unclear clauses must be resolved.
[P1-L41] “…among young children” – all young children? at which ages? please use more specific terms.
[P1-L42] “PT birth can also lead to (?) less prominent impairments” – less prominent comparing to what? to the same impairments? to no impairments? – Please revise.
[P3-LL111-113] “Participants consisted of PT individuals or individuals born with LBW, VLBW, or ELBW, both across various age ranges (infants to adults).” – Individuals initially born prematurely? Individuals initially born with LBW, VLBW, or ELBW? Please clarify the sentence.
[P3-LL141-145] “have been shown (what?) between males versus females” – Please check the logic of the sentence. Please improve. If needed, please split up to two sentences.
[P4-L159] “according to imaging approach” – imaging modality? Please clarify.
[P4-LL167-168] “Four articles were excluded as (? one specific article ?) the article was not accessible for one abstract and three other abstracts” – Please clarify the sentence.
[P4-L178] “In general, the quality of the search results was (? estimated as) satisfactory.” - Please clarify.
[P4-L182] “three studies did not exclude participants with a medical history of brain lesions” – they did not mention this point in their papers? Or they directly indicated on inclusion of such cases? Please clarify.
Please check the brain-development-concerning storyline in the sentences [P2-LL84-96]. In this version, sentences are a bit salutatory logically.
[P2-L74] “invariable facial features” – invariant? Please clarify.
[L2]
Other minor suggestions.
Please avoid starting sentences with digits like in [P1-L32]. Please use word instead of digits.
[P1-L44] “for developing (subclinical) symptoms” – since symptoms are not syndromes, it is possible to dismiss the word “subclinical”.
Reviewer 2 Report
Comments and Suggestions for Authors
The paper focuses primarily on the systematic review of face processing in premature infants, but it also mixes other related aspects into the discussion, particularly emotions and neurophysiological data.
Some concerns require revision to make the paper scientifically sound. Due to the high interest in this kind of review for the state of the art, and the (research and ethical) impact this paper could have, I strongly suggest to fix all the following issues before resubmitting the paper, to make it not only more sound, but also more clear, transparent about context-related issues, and useful for future research, where especially computer scientists and engineers, who are not experts in the context, can use advanced techniques to improve the state of the art without introducing critical biases or errors.
1) Based on current scientific knowledge, no single neuroimaging modality (MRI, fMRI, EEG, MEG) or other set of physiological data can be perfectly and exclusively related to specific emotions without some degree of overlap. Each provides valuable insights into brain activity and structures related to emotional processing, but each has limitations in terms of specificity and exclusivity. This should be clearly stated in the dedicated Limitations section and introduced in the Introduction to avoid readers accepting some questionable background knowledge as true.
2) Bias and ethics are only partially disclosed.... Data may be biased or unbalanced not only because of the lack of consistent definitions of prematurity across studies. They can be biased by ethnicity, physical disability (or lack thereof), biological sex (for data representativeness, differences in facial expression, different neural patterns in neuroimaging), and other issues that might influence the results.
This needs to be stated in the limitations.
3) It seems that in some points the paper confuses the examination of facial expressions with emotions. While the former may convey some elements that make the latter seem trivial, they are not the same. Facial expression is only part of emotional expression, and not all elements of a facial expression can be associated with emotion. In the context of premature infants, we may also have pain as a critically relevant element: pain due to therapies, due to health conditions, due to underdeveloped organs. This needs to be clearly stated in the limitations as well as defined in the introduction, and references to pain assessment in infants should be integrated into the state of the art. There are many different pain scales used in preterm infants; it is not necessary to cite them all, but some review of them or some relevant paper must be cited to give the reader a fuller idea of the context. In addition, there is at least one scientific paper that describes a protocol for automated detection of pain in preterm NICU patients: Recognizing and Predicting Neonatal Pain in Preterm Intensive Care Unit: a Study Protocol. WI/IAT 2022. This paper already includes relevant work that could be included in the state of the art, besides being a critical resource for this work, also being a seminal paper on the topic. The pain element cannot be ignored: it must be included for each analyzed paper that mentions it, and if this information is not given in the analyzed papers, it should be considered anyway to give a better idea of the context. Moreover, pain can be directly compared to emotions, both from a neurological point of view (e.g., general activation of the ACC, insula, amygdala, activation patterns of the PFC) and from a behavioral point of view (expressions: wrinkles, grimaces; body movements) and from a duration point of view.
Moreover, the fact that newborns (especially premature infants) have not yet developed long-term sensitization or a protective system such as nociceptive modulation makes pain more critical than what we may (or may not) call "emotion" at this stage of development.
4) In the context of premature infants, we can't talk about emotions the way we do with pediatric patients. They haven't developed enough of the physiology and neurology of the neural "model" of emotions. This concept can be better explained, and the model of emotion used by each paper included in the review should be included in a table. Mixing different models of emotions makes the reviewed papers not directly comparable.
5) Behavioral and physiological data, in addition to what is already mentioned in point 3, should not be mixed, but should be carefully noted for each paper analyzed, so as not to make the reader think that they are on the same level, when they refer to different contexts and approaches.
6) The paper includes studies that may include neonates from both NICU and other hospital or home environments, without detailed distinction. The differences would critically affect the results, not only in terms of early life stress, care, and sensory stimulation, but also in terms of facial recognition. Newborns in the NICU are likely to have pain affecting their physiology and neurological patterns, and they are likely to have many elements such as tubes (e.g., parenteral nutrition), electrodes, oxygen delivery, and various medical devices (very complex in the NICU) and tapes used to stick them to the body, which partially obstruct the image. The same is true for images taken with the incubator in the image vs. those taken without it. This should also be taken into account in the analysis as well as in the limitations regarding the possibility of a direct comparison between the studies, since it can influence the analysis of facial expressions, behavioral observations and other data.
7) The paper does not provide sufficient detail on how disagreements between authors were resolved during the data selection process. Adding more information on this aspect, such as the protocol used (e.g., third party arbitration or consensus meetings), whenever this information is available, would increase the transparency and rigor of the review and highlight which papers are stronger or weaker in their findings.
8) The presentation of results could benefit from additional figures or summary graphs to compare results across developmental stages or to highlight key findings.
9) Including a summary of key findings at the end of each section would improve readability.
10) The conclusion that birth weight is more critical than gestational age for face processing or developmental outcomes may be inconclusive because weight and gestational age are closely related and not entirely independent variables. This should be taken into account.
Comments on the Quality of English LanguageWhile the English is understandable, there are areas where clarity and readability could be improved. I suggest revising some of the complex sentence structures and improving grammatical consistency to improve the readability of the paper. Clearer explanations will help convey complex concepts more effectively, especially for non-expert (or interdisciplinary) readers.
Round 2
Reviewer 1 Report
Comments and Suggestions for Authors
The authors improved the manuscript extensively, solving most points had been arisen previously.
Some improvements still are required only.
[1]
Please improve the abstract
[P1-L18] “an overview of 27 studies between 2000 and mid-2022”
[2]
To improve clarity on papers analysed, please provide future readers with an additional figure depicting person ages or age intervals analysed in those papers as end points.
At this novel figure, please use the x-axis as an age axis for individuals studied in papers reviewed, depicting studied ages analysed for behavioural and functional studies separately.
[2a]
After completing the figure, please also state in the text of the manuscript whether any age intervals remained under-analysed in terms of behavioural or functional studies in persons born prematurely or with low weight.
[3]
Please specify the title for the subsection “3.4. Results” vs the title of the section “3. Results”.
[4]
In the limitation section, please clearly state that utilised search strategy might miss some studies in the interval between 2000 and mid-2022, which should be resolved in the future studies.
Comments on the Quality of English LanguageMinimal language corrections still are required.
[P1-L42] “Though the survival rate has improved…” – in babies? after 2000? Please improve the clause
Reviewer 2 Report
Comments and Suggestions for Authors
The authors applied most of the suggested reviews, improving the manuscript's precision and quality. However, a few suggested revisions were discarded, with good argumentation provided in the authors' response letter but no change to the paper.
The general idea is that if something is not clear to the reviewer, who is usually someone working in the field, the same points are likely to be clear to readers from other disciplines. Since this is a multidisciplinary paper that applies a review in several research areas, any point that might confuse the reader should be accompanied by a sentence that can clarify it for any reader.
Please, re-read all the points where an answer has been given only to the review and try to make it clear to any reader in the paper.
In particular, with regard to the recognition of facial expressions in NICU patients or preterm infants, the reviewer acknowledges that none of the studies included in the paper explicitly state that the result of the analysis was derived from the baby's facial expression. On the other hand, several studies included in the review use technologies (e.g., eye tracking) that are strongly influenced by expression. As mentioned in the previous review points, one of the main mistakes is to confuse the facial expression with the recognition of the expression.
The following points, thus, have still to be cleared up:
-In addition to specific studies, facial expression can be influenced by other factors that introduce noise, such as pain, especially in NICU patients or patients who cannot express pain verbally due to underdevelopment or disability.
- The main point is the analysis of physiological/neurological/visual data to understand the person's ability to recognize facial expressions: in this paper, many studies analyzed apply facial expressions to emotion recognition, and if they are included, the limitations or other comments about emotion recognition are still valid. Affective tasks are included in the main goal of the paper and listed among the main features of the reviewed works, so it is not clear (to the reviewers, thus potentially to any reader) why -as the authors claim in the cover letter- what has been pointed out about emotions is outside the scope of the paper. Such points should be discussed in the paper, in the subsections dedicated to general comments.
- Confusing emotions with other elements such as moods, cognitive correlated states, general health state, and others, is one of the main mistakes usually made in affective computing works, where in particular engineers and computer scientists apply algorithms that they know to application contexts that they don't know. Due to the multidisciplinary nature of this work, the authors should make every possible effort to avoid giving confused ideas of these so different elements, which are recognizable from different data, sometimes with overlapping, sometimes not, and must not be confused in the general introduction. Avoiding confusion in this work will avoid confusion in further work by other people from different disciplines applying their ideas to the same context.
